# Structural and Insulating Behaviour of High-Permittivity Binary Oxide Thin Films for Silicon Carbide and Gallium Nitride Electronic Devices

**DOI:** 10.3390/ma15030830

**Published:** 2022-01-22

**Authors:** Raffaella Lo Nigro, Patrick Fiorenza, Giuseppe Greco, Emanuela Schilirò, Fabrizio Roccaforte

**Affiliations:** Consiglio Nazionale delle Ricerche—Istituto per la Microelettronica e Microsistemi (CNR-IMM), 95121 Catania, Italy; patrick.fiorenza@imm.cnr.it (P.F.); giuseppe.greco@imm.cnr.it (G.G.); emanuela.schiliro@imm.cnr.it (E.S.); fabrizio.roccaforte@imm.cnr.it (F.R.)

**Keywords:** insulators, binary oxides, high-κ dielectrics, power electronics, wide band gap semiconductors

## Abstract

High-κ dielectrics are insulating materials with higher permittivity than silicon dioxide. These materials have already found application in microelectronics, mainly as gate insulators or passivating layers for silicon (Si) technology. However, since the last decade, the post-Si era began with the pervasive introduction of wide band gap (WBG) semiconductors, such as silicon carbide (SiC) and gallium nitride (GaN), which opened new perspectives for high-κ materials in these emerging technologies. In this context, aluminium and hafnium oxides (i.e., Al_2_O_3_, HfO_2_) and some rare earth oxides (e.g., CeO_2_, Gd_2_O_3_, Sc_2_O_3_) are promising high-κ binary oxides that can find application as gate dielectric layers in the next generation of high-power and high-frequency transistors based on SiC and GaN. This review paper gives a general overview of high-permittivity binary oxides thin films for post-Si electronic devices. In particular, focus is placed on high-κ binary oxides grown by atomic layer deposition on WBG semiconductors (silicon carbide and gallium nitride), as either amorphous or crystalline films. The impacts of deposition modes and pre- or postdeposition treatments are both discussed. Moreover, the dielectric behaviour of these films is also presented, and some examples of high-κ binary oxides applied to SiC and GaN transistors are reported. The potential advantages and the current limitations of these technologies are highlighted.

## 1. Introduction

Today, it is widely recognized that microelectronic devices have improved the quality of our daily lives, strongly contributing to the development of human civilization. In the 1940s–1950s, the first microelectronic devices appeared, and they were based on germanium. However, silicon (Si) gradually began to be the semiconductor of choice, driving the power electronics revolution with the introduction of the first p-n-p-n transistors in 1956 at Bell Laboratories [1,2]. About two decades later, the introduction of metal-oxide-semiconductor field-effect transistors (Si-MOSFETs) set the foundations for the development of the modern CMOS technology [3]. Hence, for about fifty years, microelectronics have been based mainly on Si semiconductors. The great success of digital technology may apparently indicate that Si is still the most suitable material for microelectronic devices. However, in other fields, such as electronic systems for power transmission or distribution (power converters, base stations, wireless connections, etc.) and optoelectronics (light emitting diodes—LEDs, lasers), the achievement of the ultimate silicon performances opened the route for the post-Si era. In this context, wide band gap (WBG) semiconductors emerged as the most suitable materials for this technological revolution, especially in high-power and high-frequency electronics [4,5,6,7].

Among the WBG semiconductors, silicon carbide (SiC) and gallium nitride (GaN) are the most attractive candidates because they already provide a good compromise between their theoretical properties (blocking voltage capability, operation temperature, and switching frequency) and commercial availability [4,5,6]. Their wide band gaps result in higher breakdown voltage and operation temperature with respect to Si, so both are excellent candidates to replace Si in the next generation of high-power and high-frequency electronics. Because of their different physical and electronic properties in terms of carrier mobility and thermal conductivity [8,9], SiC and GaN will cover different market segments in the post-Si technologies [10]. In particular, SiC is more suitable for high-power applications based on vertical devices, while GaN is more efficient for high-frequency applications based on lateral transistors. In any case, both materials can provide superior performances with respect to the existing Si devices [5,6], although the different technological steps for transistor fabrication need to be appropriately integrated.

Gate insulators are certainly the most important brick for transistor operation, even in the post-Si era, since the device performances critically depend on the choice of the insulating material. However, gate insulator technology is rather different in SiC and GaN, thus leading to a variety of issues to be faced when developing devices on these two WBG semiconductors.

Traditional dielectric materials, such as silicon oxide or silicon nitride, have also been widely investigated [11,12,13,14] for applications based on WBG semiconductors. However, the performance of the ideal Si/SiO_2_ system has been not achieved, and attention has been focused on the so-called “high-κ” oxides [15,16,17,18,19,20]. Among all the high-κ materials, some binary oxides (such as Al_2_O_3_ [21,22], HfO_2_ [22], NiO [23,24], CeO_2_ [25], Sc_2_O_3_ [26,27], La_2_O_3_ [28], Gd_2_O_3_ [28], Y_2_O_3_ [28,29], ZrO_2_, [17,18], Ga_2_O_3_ [30], etc.) potentially represent a suitable solution for the integration in WBG-based devices because of their higher chemical stability and/or lower fabrication cost. Some other possible materials have been studied, such as ternary oxides and nitrides, but those materials are beyond the topic of this review paper.

Table 1 shows a summary of the possible oxide candidates for the replacement of the SiO_2_ dielectric material and their principal physical properties, such as dielectric constant values, band gaps, and crystallization temperatures.

Figure 1a reports the values of the band gaps of different insulators as a function of their relative permittivity (in units of the vacuum permittivity ε_0_). The general trend (highlighted by the continuous line) is a decrease in the band gap with increasing permittivity. Hence, the reduced band gap of high-permittivity oxides can represent a concern in terms of leakage current. For this reason, insulators with appropriate band alignment with the semiconductor must be preferred. In this context, Figure 1b shows the band alignment of several high-κ oxides with the semiconductor materials under consideration (i.e., Si, 4H-SiC, and GaN). The offset between the conduction bands of the semiconductors and insulators is reported in scale.

Hence, in terms of physical properties, the guidelines for the choice of the ideal gate dielectric material are: (i) high dielectric constant value; (ii) appropriate alignment of the band gap with respect to the substrates (in particular, the band offset should be greater than 1 eV); (iii) thermal stability during the fabrication process (many steps have to be carried out at high temperatures for short periods of time) [17,18,19].

Moreover, since the gate oxide is directly in contact with the device channel, another important requirement is good quality of the gate oxide/semiconductor interface in terms of low roughness and low density of electronic defects [5].

These requirements could be met throughout two possible approaches, i.e., a crystalline gate oxide epitaxially grown on the semiconducting substrate or an amorphous oxide. Electronic defects can be thus minimized either by exactly or randomly saturating the dangling bonds, respectively. Generally, amorphous oxides are the preferred solution, since they possess isotropic dielectric constants due to the fluctuation of the polarized bonds and do not possess rough edges. By contrast, the advantage of the epitaxial oxides is the abruptness of the interface [17,18].

In general, as schematically illustrated in Figure 2, structural and compositional defects of binary oxides (e.g., oxygen vacancies, impurities, etc.) can generate the presence of energetic levels within the band gap or at the interface, and the trapped charges in these states are undesirable for the following reasons: (i) they are responsible for a shift in the voltage threshold of the transistor; (ii) they may change over time and determine the instability of the transistor output characteristics; (iii) they scatter the carriers in the inversion channel and, consequently, limit the channel mobility; (iv) they compromise the transistor reliability because they are the main cause of the dielectric breakdown [17,18].

Silicon dioxide (SiO_2_) [15] was considered an ideal dielectric during the Si era because it possesses a very low electronic defect density. The reason for this is the low coordination number, which guarantees the possibility to “repair” the dangling bonds. On the other hand, alternative high-κ oxides possess chemical bonds that cannot easily relax, thus inevitability leading to a higher electronic defect density. Hence, there is a need to reduce the number of electronic defects in these materials by annealing treatments or by optimizing their deposition processes.

In this context, the important role of the growth technique for the deposition of the high-κ dielectric layers is clear. Certainly, many deposition techniques based on either physical or chemical principles are available. However, the semiconductor industry currently demands manufacturing techniques able to achieve good surface coverage on large areas, high conformity on three-dimensional structures, high growth rate, reliability, and compatibility with the thermal budget required for the device fabrication [31,32].

Table 2 compares the main features of the common growth techniques used [29] for the deposition of high-κ oxide thin films for microelectronics applications, considering the different deposition parameters. High deposition rates and large varieties of available materials are certainly the main advantages of molecular beam epitaxy (MBE) or chemical vapor deposition (CVD) methods. By contrast, these techniques are characterized by the need for high deposition temperatures. Physical vapor deposition (PVD)-based techniques are generally preferred for metals rather than for insulator deposition and lack uniformity over large areas.

However, judging from the latest industrial trends and looking forward at the nanometric-scale miniaturization process of electronic devices, the employment of deposition methods with atomic-level accuracy has become mandatory. From this perspective, atomic layer deposition (ALD) is the most promising deposition technique, and it is gradually replacing CVD and PVD techniques in many applications.

ALD is an innovative thin-film growth method that belongs to the general class of CVD techniques. As in a typical CVD process, films are deposited from gaseous chemical precursors, one for each element of the desired compound. However, unlike the traditional CVD mechanism, the ALD process is characterized by “self-limited” reactions, first between precursor and pristine surface and second on a surface saturated by one “monolayer” of precursor fragments [31]. This deposition mechanism allows subnanometer control of film thickness, conformal coating of nonplanar substrates (step coverage ~100%), and high-quality films deposited at relatively low temperatures [32]. For these reasons, the employment of ALD can give several advantages over that of either CVD or PVD. Finally, the low growth rate of the classical thermal ALD (T-ALD) process has been now significantly improved by the implementation of plasma enhanced ALD (PE-ALD). PE-ALD is an energy-enhanced deposition technique based on plasma ignition to enhance the co-reactants’ reactivity. The high reactivity of the plasma species produces a higher density of reactive surface sites. Consequently, higher growth rates and better properties of the resulting films in terms of density, impurity content, and electrical parameters can be obtained. Another advantage of PE-ALD is the possibility to control additional process parameters, such as the operating pressure, plasma power, and plasma exposure time. Varying the plasma parameters enable fine tuning the properties of the deposited films.

A great part of the results presented in the following Sections are related to high-κ oxides grown by ALD techniques.

## 2. Amorphous High-κ Oxides on WBG Semiconductors

Several amorphous materials have been studied in the last decades as possible high-κ gate oxides for WBG semiconductors. Among them, because of their high crystallization temperature, Al_2_O_3_ thin films have certainly been the most widely investigated solution as amorphous dielectric layers. Some studies have reported on Al_2_O_3_ formed by reactive ion sputtering [33,34,35], oxidation of Al in oxygen ambient at high temperatures [36], and a few others nonconventional techniques [37,38]. The major drawbacks of these solutions are the low breakdown fields (around 5–6 MVcm^−1^) of the deposited films and their poor thickness uniformity on large areas. These limitations have been overcome by the implementation of the ALD technique, which has been the method of choice to study the potentiality of Al_2_O_3_ thin films [39,40,41,42,43,44].

However, several issues still remain objects of investigation in order to optimize the quality of deposited materials and their interfaces with the WBG semiconductors. Moreover, though the growth of high-κ oxides amorphous films is generally carried out at low deposition temperatures (in the 200–300 °C range), some interfacial interaction could occur in SiC and GaN substrates, resulting in the presence of unwanted materials or deposition by products.

In this context, the cleaning of the substrate surface before dielectric deposition, as well as the postdeposition annealing treatments, are discussed in the next subsections, illustrating as examples some relevant case studies of amorphous high-κ oxides on SiC and GaN substrates.

### 2.1. Growth of Amorphous High-κ Oxides on SiC

Unlike that of thermal silicon dioxide (SiO_2_), the growth of high-κ oxides on silicon carbide is much more affected by the quality of the semiconductor surface. In fact, in order to limit the amount of the interface state density (D_it_), appropriate cleaning of the SiC surfaces is always required.

A variety of SiC surface-cleaning treatments have been proposed, based either on wet chemical solutions [44,45,46] or plasma [47,48,49]. The most used chemical solutions for SiC cleaning are combinations of diluted sulfuric acid, hydrogen peroxide, isopropanol, diluted hydrofluoric acid. Suvanam et al. [46] demonstrated that RCA treatment [45], followed by HF diluted solution and finally isopropanol, was a good route to improve the interfacial electrical characteristics of Al_2_O_3_ films on SiC, obtaining a density of interface states D_it_ = 1.5 × 10^11^ eV^−1^ cm^−2^ at E_C_ − E_t_ = 0.2 eV below the 4H-SiC conduction band edge, which was about two orders of magnitude lower than the values found with thermal SiO_2_. In regard to plasma treatment before high-κ deposition, H_2_ plasma has been also evaluated in some works [47,48,49], since it represents an efficient route for the passivation of dangling bonds on SiC surfaces. Heo et al. [49] measured promising values of interface state density (D_it_ = 6 × 10^12^ eV^−1^ cm^−2^ at E_C_ − E_t_ = 0.2 eV) when a 15 min long H_2_ plasma treatment was performed before deposition and after the post-metallization step.

As a matter of fact, besides surface treatments before the dielectric deposition, postdeposition annealings are of great importance to optimize the dielectric properties. Many parameters can in principle be varied, such as ambient atmosphere, annealing temperature and time, etc. However, these processing steps must be ultimately compatible with complete SiC device fabrication, in which, e.g., the formation of metal contacts is achieved at high temperatures (900–1000 °C) and fixed gas atmospheres (N_2_ or Ar). Generally, a large number of high-κ oxides possess crystallization temperatures of about 400–500 °C, with Al_2_O_3_ being the most thermally stable at up to 800 °C. However, independently of the chemical nature of the high-κ oxide, the annealing process can improve dielectrical properties. For instance, Wang et al. [50] demonstrated the beneficial effects of high-temperature annealings (800–1000 °C) performed in O_2_ atmosphere on Al_2_O_3_ films. In particular, they showed that although Al_2_O_3_ films started crystallizing at 900 °C, capacitance vs. voltage (C–V) measurements revealed their improved electrical characteristics (i.e., reduced hysteresis phenomena). Hence, the authors concluded that annealing at 900 °C represented the best option in terms of both surface morphology and dielectric quality. On the other hand, many other papers demonstrated that such high annealing temperatures induce the formation of a thin stoichiometric or sub-stoichiometric silicon oxide interfacial layer [33,50,51,52]. This oxidation phenomenon can have a detrimental impact on the properties of high-κ/SiC interfaces, including in the case of abrupt Al_2_O_3_/4H-SiC interfaces obtained by ALD growth [40,53,54,55]. In this context, annealing in N_2_ atmosphere can be the preferred solution, although uncontrolled SiO_x_ formation can occur in N_2_ atmosphere for high annealing temperatures. Moreover, Avice et al. [42] and Khosa et al. [36] showed that an additional effect of incomplete SiC oxidation was the formation of C clusters if not enough oxygen was present to enable the out-diffusion of carbon as carbon monoxide. The formation of the SiOx interfacial layers was observed independently of the annealing temperature or ambient. In fact, this phenomenon has been observed even in vacuum or at only 300 °C annealing temperature [55]. Hence, it is expected that the elimination of residual O_2_ molecules in the annealing ambient is one the key issues for the limitation of SiOx formation.

In general, most of the reported postdeposition annealing studies were carried out in oxidizing (O_2_ or N_2_O) or non-oxidizing (Ar, N_2_ or forming gas) ambient, in the 500–1100 °C temperature range, and for short (1 min) or long (1–2 h) times. An interaction at the interface has always been observed by the formation of the silicon oxide layers and carbon clusters. The control of the chemical nature of the interface products, which in turn strongly affects the electrical characteristics, is not trivial.

In this context, Schilirò et al. [39,40] reported an interesting comparison between the properties of Al_2_O_3_ thin films grown by PE-ALD on bare 4H-SiC and on a 5 nm thermal SiO_2_/SiC stack. TEM analyses (shown in Figure 3a,b) showed uniform interfaces and well adherent films. The surface morphology of the films (determined by AFM) was very similar, with root-mean-square (RMS) values measured over a 1 μm^2^ area of 0.670 nm and 0.561 nm for Al_2_O_3_/SiC and Al_2_O_3_/SiO_2_/SiC samples, respectively.

Though the interface structural quality appears analogous, quite different electrical properties were measured on MOS capacitors. In fact, current vs. voltage (I–V) measurements (Figure 3d) showed a higher leakage current in the Al_2_O_3_/SiC than in the Al_2_O_3_/SiO_2_/SiC stack. Furthermore, the breakdown fields, i.e., 5.7 MV/cm for the Al_2_O_3_/SiC and 7 MV/cm for the Al_2_O_3_/SiO_2_/SiC, demonstrated the better electrical quality obtained by the introduction of the SiO_2_ at the interface. Moreover, the relative permittivity values, evaluated from the C–V curves (Figure 3c), were ε ≈ 6.7 and ε ≈ 8.4 for the Al_2_O_3_/SiC and the Al_2_O_3_/SiO_2_/SiC samples, respectively.

These results can be explained by considering both the larger conduction band offset between the SiO_2_ and the SiC substrate (Figure 1b) and the different chemical impact of the substrate surface on the Al_2_O_3_ nucleation process. This latter is schematically depicted in Figure 4, showing that the presence of the OH species on the SiO_2_ surface favours the nucleation process by increasing the number of nucleation sites and the formation of denser Al_2_O_3_ films.

Other high-κ oxides have been also grown on SiC substrates as thin amorphous films, such as HfO_2_ [56,57,58], La_2_O_3_ [59,60], Ta_2_O_5_ [61], and TiO_2_ [62]. Among these materials, HfO_2_ thin films have been widely investigated because of their superior theoretical properties, such as much higher permittivity. However, the main drawback for their implementation on SiC-based devices is the imperfect alignment of both conduction and valence band offsets (about 0.7 and 1.74 eV, respectively) with those of SiC. Cheong et al. [56,57] reported on HfO_2_ films with a very high dielectric constant value (20), but the interface state density D_it_ was as high as 2 × 10^13^ eV^−1^ cm^−2^, which give no advantage with respect to the SiO_2_/SiC system. Moreover, very high leakage current densities of 1 mA cm^−2^ were already recorded in an electric field as low as 0.3 MVcm^−1^ by Afanas’ev et al. [58]. While in this case, the high leakage current could in principle be mitigated by the introduction of a SiO_2_ layer at the SiC interfaces, a further issue to be considered is the low thermal stability of HfO_2_ at temperatures higher than 500 °C, when crystallization starts to occur.

In order to maintain the best features of HfO_2_ (i.e., high permittivity) and Al_2_O_3_ (i.e., high crystallization temperature), these two materials have been evaluated in combined laminated systems.

In this context, some Al_2_O_3_/HfO_2_ bilayer systems deposited on thermally oxidized 4H-SiC substrate have been studied, the most complex stack being an Al_2_O_3_/HfO_2_ multilayer laminated system [63]. The Al_2_O_3_/HfO_2_ nanolaminate shown (Figure 5a) had a total thickness of 38 nm and perfectly distinguishable sublayers, each with thickness of about 1.4–1.8 nm. After annealing treatment at 800 °C in N_2_ atmosphere, the interfaces between the sublayers (Figure 5b) became less sharp, and an intermixing process occurred. Notably, both the as-deposited and annealed samples showed amorphous structures. AFM investigation pointed to a smooth surface morphology with a low RMS value of 0.6 nm, which was maintained in the annealed sample. A dielectric constant value of 12.4 was determined by the accumulation capacitance in MOS capacitors, taking into account of the SiO_2_ interfacial layer. However, on the as-deposited sample, a high value of oxide trapped charge (N_ot_) of 2.7 × 10^12^ cm^−2^ was found. Nevertheless, after the annealing treatment at 800 °C in N_2_, the nanolaminated stack showed an improvement of the dielectric properties, since the dielectric constant value increased to 13.4 and the N_ot_ value decreased to 1.15 × 10^12^ cm^−2^.

Few other papers have been dedicated to thin films of simple high-κ oxides such as La_2_O_3_ [59,60], Ta_2_O_5_ [61], or TiO_2_ [62], which, when directly grown on 4H-SiC, showed analogous results as in the case of simple HfO_2_ oxide. Generally, they demonstrated good dielectric constant values, but their high interface state density and low breakdown voltages made them still far from possible implementation in real devices.

In summary, among the pure high-κ oxides, Al_2_O_3_ thin films represent the best compromise, especially in combination with a very thin SiO_2_ interfacial layer. Some possible other high-κ bilayers, such as HfO_2_/Al_2_O_3_ [64], Y_2_O_3_/Al_2_O_3_ [65], or ZrO_2_/SiO_2_ [66], exhibited some potentiality, although not many reports have been made available to date, especially regarding devices.

In regard to dielectric properties, the relevant results on the electrical performances of high-κ oxides integrated in SiC MOSFETs are reported in more detail in Section 4.

### 2.2. Growth of Amorphous High-κ Oxides on GaN-based Materials

The surfaces of GaN-based materials (GaN, AlGaN, InGaN, etc.) are typically characterized by the presence of large concentrations of defects (e.g., nitrogen vacancies, structural/morphological imperfections, residual contaminations, etc.) that can result in large leakage current and low performance and device reliability. Kerr et al. [67] demonstrated by density functional theory simulations that the defect sites, such as Ga dangling bonds and Ga-Al metal bonds, are responsible for the formation of states in the band gap. These interfacial trap states could be removed by annealing procedures before or after gate dielectric deposition. Moreover, especially from the perspective of high-κ gate oxide deposition, the removal of contaminations is crucial for increasing the density of precursor nucleation sites. Hence, pre-deposition surface treatments are needed to improve high-κ oxide quality. Systematic studies [68,69,70,71,72,73,74,75,76] have reported on the effect of several pre-treatments, and the principal cleaning/activation methods have been based on the use of wet chemical solutions [68,69,70,71,72,77,78] or plasma/gas actions [73,74,75,76]. Generally, the piranha (H_2_O_2_:H_2_SO_4_) solution is used for the cleaning of carbon contaminations, but some oxidation of the nitride surface can occur [70,71]. On the other hand, chloride acid (HCl) solution is efficient for the removal of metallic contaminations (eventually present from device processing) or residual oxygen on the surface. However, chlorine itself could be a residual contamination of the system [70]. Finally, hydrofluoric acid (HF) treatment is effective for the elimination of unwanted native oxide formation but is not efficient for carbon contamination [70,71]. Brennan et al. [71] compared the nucleation efficiency of the Al precursor with/without the cleaning of the surface by sequential use of acetone, methanol, isopropanol, and HF 2% solution. It was clear, from the results of an XPS study after each ALD cycle, that the decrease in the Ga-O concentration induced by the HF etch resulted in a stronger interaction between the Al precursor and the Ga surface. Nepal et al. [69] compared the effects of three different chemical solutions (i.e., piranha, diluted HF, and diluted HCl), finding that: (i) the single HCl pre-treatment provides 10–30 nm-sized particles, indicating a three-dimensional nucleation; (ii) the HF-based treatments produced an improvement in the electrical behaviour; (iii) the best dielectric properties, in terms of smaller hysteresis and lower density-trap state values, were obtained on the piranha-treated surface. Finally, Schilirò et al. [72] showed a comparison among several chemical solution combinations (i.e., piranha, HCl/ HF, and piranha/HF). In particular, it was shown that, although the Al_2_O_3_ thin films treated with each solution possessed identical structural properties, adherent, uniform, and amorphous, there were some intrinsic differences depending on the adopted surface pre-cleaning. In fact, under a TEM electron beam, the films deposited after piranha treatment showed the formation of polycrystalline grains, while epitaxial layers were formed for samples deposited after HF based treatments. This was an indication that in the case of HF-based treatments, the deposition process occurred on a very clean AlGaN surface, which could act as seed layer for the formation of epitaxial films. Moreover, investigation of the initial growth stages by AFM demonstrated that the smallest three-dimensional grain nucleation resulted in deposition on HF-HCl-treated surfaces, which could ensure a cleaner surface in order to allow ideal layer-by-layer ALD growth.

It could be concluded that the pre-deposition treatments of GaN-based surfaces with HF cleaning provided Al_2_O_3_ films with the best dielectric properties [69,71,72].

An alternative route to cleaning by chemical solution is represented by “in situ” cleaning process based on H_2_/N_2_ (forming gas) or NH_3_ plasma actions [68]. The impact of N_2_ and forming gas on the growth and interfacial characteristics of Al_2_O_3_ on AlGaN/GaN heterostructures was explored by Qin et al. [73], who demonstrated by XPS investigation that C contamination was effectively reduced by both N_2_ and forming gas plasma. The latter also decreased the number of Ga-O bonds, improving the Al_2_O_3_ nucleation. In regard to plasma action effects before high-κ deposition, the same group contributed with a large variety of studies [73,75,76]. In particular, the effects of O_2_, N_2_, and forming gas plasma annealing were evaluated, comparing the electrical behaviour in terms of interface state density with the results obtained by XPS analyses. The formation of oxynitride bonds (Ga-O-N) increased the number of interface defects and that among all the studied treatments, the forming gas action was the most efficient.

In this context, it has to be emphasized that the semiconductor surface preparation and the deposition conditions may induce different insulting behaviours after the first film growth stages. As an example, Schilirò et al. [79] recently reported different behaviour in the early growth stages of Al_2_O_3_ thin films deposited on AlGaN/GaN heterostructures by thermal or plasma-enhanced ALD. In particular, they provided evidence that the PE-ALD process occurred under ideal layer-by-layer growth because of the efficiency of the O_2_-plasma agent, which acted directly on the Al precursor. On the other hand, the T-ALD approach resulted in a nucleation process of the Al_2_O_3_ film similar to the island-growth model and a higher susceptibility to charge trapping [79].

Summarizing, surface preparation prior to high-κ oxide deposition is a crucial issue, including in the case of GaN-based materials, and can be carried out by many procedures. The aim is the cleaning of C residues, which are detrimental for the oxides’ nucleation, and the elimination of Ga-N-O bonds, which are the main centres of interfacial electronic defects. These two issues are generally addressed by non-oxidizing plasma action or by HF treatments.

## 3. Epitaxial Growth of High-κ Oxides on WBG Semiconductors

While different oxides have been studied as gate insulators on SiC and GaN [15,16,17,18,19], only some of them can be grown epitaxially on the WBG semiconductor single-crystal surface [23,24,25,80,81,82,83,84]. The epitaxial growth of high-κ oxides on WBG semiconductor substrates can offer some advantages. Generally, the principal improvement is related to better saturation of interface unbonded atoms. In particular, the most commonly used SiC and GaN polymorphs for microelectronics applications possess the wurtzite structure, with hexagonal surface atomic arrangements. However, though in principle this strategy can be applied to both SiC and GaN technologies, practical studies have been performed mainly on GaN-based substrates. In fact, the few studies of epitaxial high-κ materials on SiC substrates were limited to γ-Al_2_O_3_ phase films [80] and direct growth of NiO thin films by metal organic chemical vapour deposition (MOCVD) [81]. The γ-Al_2_O_3_ phase films were initially grown by Tanner et al. [80] by the ALD process as amorphous layers, and the epitaxy on 4H-SiC substrate was obtained under a post-annealing crystallization process at a very high (1100 °C) temperature. The epitaxy was observed for the alignment of the γ-Al_2_O_3_ (111) planes with the (001) 4H-SiC substrate, having a lattice mismatch of about 8.8%. On the basis of the performed reflection high-energy electron diffraction analysis, the (111) γ-Al_2_O_3_ oriented films showed quite good structural properties for film thickness up to 20 nm, even though some twinned grains were present. Moreover, upon increasing the film thickness, the crystallization process was no more efficient, and amorphous regions were observed under TEM investigation.

Epitaxial NiO films, by contrast, have been directly grown onto 4H-SiC epilayers at the deposition temperature of 550 °C [81,85]. A high-resolution TEM micrograph of the NiO/4H-SiC interface (Figure 6a) confirmed the presence of an axially-oriented (111) NiO film, but a “non-ideal” interface was observed, because a discontinuous amorphous SiO_2_ layer was detected, probably formed during MOCVD growth. Furthermore, the presence of Moiré fringes generated by the superposition of twinned NiO grains was observed. The C–V characteristics of NiO/4H-SiC capacitors (Figure 6b) were used to calculate the dielectric constant, the value of which, at 6.2, was much lower than the theoretical 11.9. This result was justified by the presence of the discontinuous silicon oxide interfacial layer.

More studies on growing epitaxial oxides have been carried out on GaN-based materials. The materials under investigation comprise some lanthanide oxides, such as gadolinium [82], scandium [83,84], and lanthanum [83] oxides, as well as nickel [23,24,81] and cerium oxides [25,81]. The lanthanides oxides possess bixbyite symmetry, while NiO and CeO_2_ are face cubic centred (fcc) oxides. However, the (111) planes of the latter two possess a hexagonal oxygen structure, which is suitable for epitaxy with the (0001) GaN superficial planes. Their structural and physical properties are summarized in Table 3.

The epitaxial growth of Sc_2_O_3_ thin films was performed on a GaN substrate at about 700 °C by the pulsed laser deposition (PLD) technique [84]. Herrero et al. [84] demonstrated that the most critical deposition parameter to obtain perfectly stoichiometric and epitaxial Sc_2_O_3_ thin films was the oxygen partial pressure. In particular, above 50 millitorr oxygen partial pressure, more than one preferential growth direction was observed. The epitaxial growth of Sc_2_O_3_ was also evaluated by Jur et al. [83] by the MBE technique. Their investigation extended to La_2_O_3_, which in principle can provide a dielectric constant of 26 in its hexagonal structure. Nevertheless, La_2_O_3_ growth was demonstrated not to be trivial, since La_2_O_3_ is a hygroscopic material and tends to form an amorphous layer at the interface with the GaN substrates. Nevertheless, the authors demonstrated that it was possible to inhibit the water diffusion by the introduction of a thin Sc_2_O_3_ layer between GaN and the growing La_2_O_3_ films. The MBE technique was also used for the growth of Gd_2_O_3_ epitaxial gate oxide on an AlGaN/GaN heterostructure [82]. Sakar et al. [82] showed the impact of a Gd_2_O_3_ epitaxial oxide layer on the electrical performance of an HEMT device. Gd_2_O_3_ films were deposited at 650 °C. The authors demonstrated that the Gd_2_O_3_ layer underwent phase transition upon increasing its film thickness. The first layers, up to about 3 nm, possessed hexagonal structure, which changed to monoclinic phase when the thickness of 15 nm was reached. This phase transformation had a great impact on the electrical properties, especially in terms of interface trap density, which showed a minimum value of 2.98 × 10^12^ cm^−2^ eV^−1^ in Gd_2_O_3_ film 2.8 nm thick. The authors’ conclusion was that the epitaxial lattice strain also positively affected the two-dimensional electron gas density at the AlGaN/GaN interface by about 40%.

Nickel and cerium oxides (NiO and CeO_2_) have also been deposited onto AlGaN/GaN systems. The first report on NiO-oriented film as a gate insulating layer in AlGaN/GaN devices was related to thermal oxidation of Ni metal layers [86]. In particular, the fabrication process relied on a heating treatment, in the 300–600 °C temperature range for 5 min in air ambient, of a 10 nm-thick Ni metal layer. Besides the observation of a colour change from the dark Ni metal layer to the transparent NiO film, no details were provided on the structural or compositional characteristics of the formed NiO layers. Generally, the thermal oxidation of Ni metal layers can lead to the formation of voids in the oxide layer and/or of randomly oriented films, since the process initiates at the grain boundaries and then expands in all directions. The growth kinetics of NiO film seem to depend on the texture and crystallite size of the initial Ni metallic layer [87]. It has been shown that the strong (111) texture of the Ni layer results in slow NiO growth. These slow oxidation kinetics are related to the stronger resistance to oxidation of the Ni (111) planes [88]. Therefore, the NiO growth proceeds mainly from other crystallographic planes, mostly located at the grain boundaries. Indeed, most of the Ni grains have a (111) texture. This nonuniform growth results in increased surface roughness after oxidation.

The growth of NiO and CeO_2_ thin films on AlGaN/GaN heterostructures was carried out by MOCVD at 500 °C [23,24,25]. TEM analysis demonstrated the formation of 16 nm-thick NiO (Figure 7a,b) and 20 nm-thick CeO_2_ (Figure 7c,d), both compact and uniform films. Since no intermediate layers were visible at the interface, the occurrence of any interaction and/or oxidation of the substrate during the growth process was ruled out. Moreover, the (111) NiO planes were perfectly parallel to the (0001) planes of the AlGaN/GaN substrate.

The selected area electron diffraction (SAED) pattern (Figure 7b) indicated that the external spots related to the NiO were perfectly aligned to the internal ones from the AlGaN. In particular, the white spots at 2.77 Å and 1.59 Å plane distances could be related to the (100) and (110) AlGaN/GaN planes and represented the typical 0001 zone axis pattern for a hexagonal single crystal, while the red spots forming the hexagonal pattern at 1.47 Å can be related to the (220) NiO plane; thus, only the 111 NiO zone axis is visible. The NiO spots are perfectly aligned to the AlGaN/GaN spots at 1.59 Å. Hence, it is possible to conclude that an epitaxial growth of the (111) NiO planes on the (0001) substrate plane occurred. The occurrence of the epitaxial growth can be explained by considering the threefold symmetry of the (111) NiO, which makes possible an epitaxial relationship between the hexagonal (0001) planes from the AlGaN substrate and the (111) planes of the NiO film. In particular, the lattice mismatch between the two hexagonal arrangements from the NiO and AlGaN, calculated from the electron diffraction images, was about 5%. Moreover, it is worth noting that the XRD peak position of the NiO (111) reflection was very close to that of bulk NiO, thus indicating that relaxed NiO thin films with strong diffraction intensity could be obtained under the described operating conditions. Hence, it can be concluded that NiO deposited samples were epitaxial and stress-free films and possessed excellent interface quality. TEM analysis also defined the structural relationship between the deposited CeO_2_ films and the AlGaN/GaN substrate. A TEM cross-section image showed the formation of 20 nm-thick CeO_2_ film and an almost perfect film/substrate interface (Figure 7c). The presence of differently oriented grains is evident, as can be deduced by the appearance of Moiré fringes. In-plane SAED was also recorded, and diffraction patterns of three different zone axes were visible. The 0001 zone axis pattern of the substrate is represented by the white circles in Figure 7d. The CeO_2_ SAED pattern demonstrated that the CeO_2_ film grew along two different orientations, namely, the (111) and (100) directions. In fact, the 111 zone axis pattern is represented by the red spots lying at 1.93 Å plane distances, and the 100 zone axis pattern is represented by dots lying at the vertex and at the centre of each side of the yellow squares at 1.93 Å and 2.70 Å plane distances, respectively. The 100 CeO_2_ zone axis is represented by three equivalent configurations 30° rotated in the plane.

Hence, the NiO films (111) epitaxially grew on (0001) AlGaN/GaN substrate, while the CeO_2_ film was not a single crystal epitaxial layer but formed by two sets of differently oriented grains (namely, (111)-oriented and (100)-oriented grains) aligned in the (0001) substrate plane of AlGaN.

The electrical characteristics of the oriented NiO and CeO_2_ thin films allowed determining their experimental permittivity values. In fact, from the analysis of the C–V curves, it was possible to estimate permittivity values of 11.7 and 26 for NiO and CeO_2_ films, respectively. These values were very close to those of the NiO and CeO_2_ bulk permittivity (11.9 and 26) and properly higher than that of AlGaN alloys. These good values were probably due the oriented growth of the two films, which represented almost an “ideal” bulk system, in contrast to amorphous and/or polycrystalline films, which generally show lower values with respect the bulk materials.

Another key parameter to be considered in dielectric material integration onto WBG semiconductors is the effective density of the trapping states. The maximum of the trapping states determined in the AlGaN/GaN metal insulator semiconductor (MIS) diodes were 5 × 10^12^ cm^−2^eV^−^^1^ for the CeO_2_ films and 6 × 10^11^ cm^−2^eV^−^^1^ for the NiO films. The trapping states of the CeO_2_ were higher than those of the NiO, which could be attributed to the better structural characteristics of the NiO/AlGaN interface. While (111) NiO thin insulating layers seem to be an appealing choice as an epitaxial gate oxide, their integration into a real transistor has not been attempted yet.

## 4. Application of High-κ Oxides as Gate Dielectrics in SiC and GaN Transistors

As already mentioned in the introduction, most powered electronic devices based on silicon have used silicon dioxide (SiO_2_) as a gate dielectric. However, the use of SiO_2_ in modern devices based on WBG semiconductors can be a bottleneck for the full exploitation of the intrinsic properties of these materials because of the low value of the dielectric permittivity of SiO_2_.

Figure 8 shows the schematics of common insulated gate transistors based on wide band gap semiconductors (SiC and GaN), i.e., a 4H-SiC metal oxide semiconductor field effect transistor (MOSFET) (Figure 8a), an AlGaN/GaN metal insulator semiconductor high electron mobility transistor (MISHEMT) (Figure 8b), and a recessed gate hybrid AlGaN/GaN MISHEMT (Figure 8c).

A first advantage of using a high-κ dielectric in a power device is related to the distribution of the electric field at the gate dielectric region. In particular, according to Gauss’s law, the electric field in a gate dielectric E_ins_ that is placed on a semiconductor substrate, e.g., in the gate of a transistor, is given as:(1)$$E_{\mathrm{ins}}=\frac{\kappa_{s}}{\kappa_{\mathrm{ins}}}E_{s}$$
where κ_s_ and κ_ins_ are the relative dielectric permittivity values of the semiconductor and insulator, respectively, and E_s_ is the electric field in the semiconductor [89].

Considering as an example that the relative dielectric permittivity of 4H-SiC is 9.7 while that for SiO_2_ is 3.9, according to Equation (1), the electric field in the gate oxide is about a factor of 2.5 times that in the semiconductor. Hence, when the critical electric field of 4H-SiC is reached, the maximum electric field in the oxide exceeds 9 MV/cm, thus meaning that the insulator is subjected to a significant stress, and the device reliability is penalized. In recognition of this problem, it has been proposed to replace the conventional SiO_2_ gate dielectric by a high-κ insulator, with a permittivity comparable to that of SiC, so that the electric field in the gate dielectric would become closer to that in the semiconductor. In this way, the maximum electric field in the gate dielectric could be reduced, which should be satisfactory for reliable device operation. Moreover, the changes in the electric field distribution have a strong impact on the drift layer thickness required to sustain the targeted drain bias. In fact, using a high-permittivity gate dielectric allows using the optimal semiconductor drift region for the targeted breakdown, thus minimizing the specific on-resistance of the device.

Moreover, considering always the case of a SiC MOSFET (Figure 8a), the total specific on-resistance R_on,sp_ of the device is given by the sum of different contributions [89]:(2)$$R_{\mathrm{on},\mathrm{sp}}=R_{\mathrm{ch}}+R_{a}+R_{\mathrm{JFET}}+R_{\mathrm{drift}}+R_{\mathrm{sub}}$$
where R_ch_ is the channel resistance, R_a_ is the accumulation region resistance, R_JFET_ is the resistance of the JFET region, R_drift_ is the resistance of the drift region after the current spreading from the JFET region, and R_sub_ is the resistance of the n-type doped substrate.

R_a_ and R_JFET_ can be minimized by appropriately scaling the device layout, and R_sub_ can be reduced by thinning the substrate. Hence, the control of the channel resistance contribution R_ch_ is a critical point in 4H-SiC MOSFET fabrication. In particular, the channel resistance contribution R_ch_ is given by:(3)$$R_{\mathrm{ch}}=\frac{\left(L_{\mathrm{ch}}\cdot p\right)}{\mu_{\mathrm{inv}}C_{\mathrm{ox}}\left(V_{G}-V_{\mathrm{th}}\right)}$$
where p is the pitch of the MOSFET elementary cell, L_ch_ is the channel length, µ_inv_ is the mobility for electrons in the channel (inversion layer), C_ox_ is the specific capacitance of the gate oxide, V_th_ is the threshold voltage, and V_G_ is the applied gate bias. The gate oxide capacitance term C_ox_ increases with the insulator permittivity. Hence, it has a direct impact on the channel resistance and ultimately on the device’s total resistance.

As pointed out by theoretical works [89,90], the use of high-κ is ideally desirable for future application in trench MOSFET technology [91].

One of the interesting features of the GaN semiconductor and its related AlGaN alloys is the possibility of growing AlGaN/GaN heterostructures. AlGaN/GaN heterostructures are characterized by the presence of a two-dimensional electron gas (2DEG) formed at the interface and possessing a high sheet charge density (in the order of 10^13^ cm^−2^) and a high mobility (above 1000 cm^2^V^−1^s^−1^) [92,93]. Moreover, GaN-based materials have a high critical electric field (above 3 MV/cm). Thanks to these unique properties, high-electron mobility transistors (HEMTs) based on AlGaN/GaN heterostructures with excellent performances have been demonstrated in recent years and are suitable candidates for high-frequency applications [94,95]. These devices are based on a Schottky barrier at the gate electrode to modulate the channel current. However, particularly for high-voltage applications in which the gate electrode is strongly reverse biased with respect to the drain, a high gate leakage current at the Schottky junction can limit the performance of these transistors [96]. Hence, a dielectric must be introduced under the gate in order to reduce the leakage current, creating a metal–insulator–semiconductor high-electron mobility transistor (MISHEMT), as schematically shown in Figure 8b. In this case, however, the choice of the gate dielectric represents a key issue for improving device performance [21,97,98] and optimizing the parasitic capacitance and the gate leakage current [19,99].

Similarly, the benefits of using high-κ materials on the characteristics of insulated gate transistors in SiC and GaN can be understood from the theoretical calculations shown in Figure 9a,b. In particular, Figure 9a shows our calculation of the threshold voltage as a function of the thickness of different high-κ dielectrics for 4H-SiC MOSFETs. As the gate dielectric thickness is increased to reduce the gate leakage current, the threshold voltage of the device (V_th_) also increases. Hence, while an improvement in the off-state characteristics of the MOSFET is achieved, this is accompanied by a degradation in the on-state performance. However, using high-κ dielectrics as insulating gate materials instead of the conventional SiO_2_, the rate of increase in the threshold voltage with the dielectric thickness is reduced. In this way, the leakage can be reduced, with a minor side effect on the output current.

Figure 9b shows the calculation of the threshold voltage of a GaN-based MISHEMT as a function of the gate dielectric layer thickness of different high-κ insulators. In this case, the V_th_ of the device is negative because of the inherent normally-on nature of these devices [92]. The negative value of V_th_ increases with increasing thickness of the gate insulator. However, the rate of this negative shift is reduced with increasing dielectric permittivity [100]. Hence, the use of high-κ gate insulators in GaN-based MISHEMTs is beneficial for reducing the power consumption of the devices.

### 4.1. Binary High-κ Oxides in 4H-SiC MOSFETs

Since the band gap for SiC is three times larger than that for Si, the band offset at the SiO_2_/SiC interface is smaller than that in the SiO_2_/Si system. Hence, in SiC MOS-based systems, a higher tunnelling current than in Si is expected for a given oxide thickness [8].

Because of its high permittivity (20), hafnium oxide (HfO_2_) has been widely used in Si technology. Hence, this material has attracted also the attention of the SiC scientific community. In particular, the investigation started by studying the electronic structure of the HfO_2_/SiC interface [101]. However, it was clear that HfO_2_ alone is not suitable for SiC because of the low conduction band offset (in the range 0.5–0.7 eV)) at the HfO_2_/SiC interface, which may not provide an adequate barrier height for electron injection from the substrate [101,102]. Because of the intrinsic limitation of the band alignment, attention moved to the study of the HfO_2_/SiO_2_/SiC system [102].

Moreover, other high-κ binary oxides with larger band gaps and more favourable band alignment with SiC, such as Al_2_O_3_ [101], La_2_O_3_ [59,103], and ZrO_2_ [104,105], have been investigated.

In general, in order to mitigate the fundamental limitations of high-κ binary oxides, the introduction of a SiO_2_ interlayer between the high-κ material and SiC is often adopted [58,102].

A good survey of the literature on high-κ dielectrics for SiC was recently reported by Siddiqui et al. [106].

As described before, using high-κ dielectrics in 4H-SiC MOS-based devices can be beneficial to fully exploit the properties of the material and reduce the device’s on-resistance. However, combined interaction with the SiOx layer can give further improvements. As an example, high channel mobility in 4H-SiC MOSFETs with Al_2_O_3_ gate insulators fabricated at low temperatures by MOCVD (64 cm^2^V^−1^s^−1^) can be obtained when the Al_2_O_3_ gate insulator is deposited at 190 °C. According to Hino et al. [107], this result could be further improved up to an extremely high field-effect mobility of 284 cm^2^V^−1^s^−1^ when the 4H-SiC MOSFET was fabricated with an ultrathin thermally grown SiOx layer inserted between the Al_2_O_3_ and SiC interface [107].

On this particular aspect, the impact of a thin SiO_2_ layer thickness inserted between Al_2_O_3_ and SiC on the channel mobility in Al_2_O_3_/SiC MOSFETs was investigated by Hatayama et al. [108]. They demonstrated that the peak value of the field-effect mobility in Al_2_O_3_/SiO_2_/SiC MOSFETs could reach 300 cm^2^V^−1^s^−1^ for an SiO_2_ thickness of 1 nm. On the other hand, when the SiO_2_ layer increased up to 2 nm, the field-effect mobility drastically reduced to 40 cm^2^V^−1^s^−1^ [108], as illustrated in Figure 10.

Another possible approach is employing a semiconductor surface treatment prior to gate insulator deposition. Lichtenwalner et al. [43] reported the use of a NO annealing at 1175 or 1100 °C for 20 min of a 4H-SiC semiconductor in an attempt to control the interface state density D_it._ This procedure allowed obtaining a peak field-effect mobility in 4H-SiC MOSFETs of 106 cm^2^V^−1^s^−1^ using an Al_2_O_3_ film deposited by ALD as gate dielectric with postdeposition annealing at 400 °C for 30 s.

However, a key aspect is the channel mobility at the operative gate bias. In fact, the remarkable peak values of the field-effect mobility are often accompanied by a rapid decrease due to an increase in the gate bias close to the value at which the device should operate. This particular phenomenon can be understood analysing the single components limiting the channel mobility. As an example, a rapid decrease in the field-effect mobility is associated with a dominant phonon-scattering mechanism, while a smooth decrease with an increase in the gate bias is associated with coulombic scattering [109,110]. In particular, Arith et al. [111] demonstrated a process for forming aluminium oxide (by ALD) as a gate insulator in 4H-SiC MOSFET that did not involve the insertion or formation of SiO_2_ at the interface, eliminating traps that may be present in SiO_2_. This was achieved with hydrogen plasma pre-treatment followed by annealing in forming gas. Hydrogen treatment was effective at reducing D_it_ at the interface of aluminium oxide and SiC without a SiO_2_ interlayer.

Clearly, because of the large differences in the mobility behaviour of the MOSFETs processed under different conditions, this topic has been strongly debated. In particular, Yoshioka et al. [47] demonstrated optimization of the interface of aluminium oxide and SiC without a SiO_2_ interlayer, resulting in a low D_it_ for the metal oxide semiconductor (MOS) capacitor of 1.7 × 10^12^ cm^−2^eV^−1^ at E_C_ − E_t_ = 0.2 eV and a peak field-effect mobility of 57 cm^2^V^−1^s^−1^ that was quite constant with the variation of the gate bias. Other works have tried to figure out the right combination of semiconductor surface pre-treatments and postdeposition annealing in order to improve the electrical properties of Al_2_O_3_/SiC interfaces [41,46].

Other processing steps have been explored to improve the performance of 4H-SiC MOSFETs, e.g., by appropriate manipulation of the SiO_2_/SiC interface. In particular, Yang et al. [112] deposited 30 nm of SiO_2_ by ALD and subsequently performed a postdeposition annealing (PDA) in a nitrous oxide (N_2_O) ambient. The highest electron mobility of 26 cm^2^V^−1^s^−1^ was achieved by performing PDA at 1100 °C for 40 s. The gate oxide could withstand effective fields up to 6 MV/cm within a leakage current range of 1 × 10^−7^ A/cm^2^. This value of maximum electric field was small compared to that of thermally grown SiO_2_, which can typically withstand up to 10 MV/cm. In another work, Yang et al. [113] inserted 1 nm of lanthanum silicate (LaSiOx) between ALD-deposited SiO_2_ and 4*H*-SiC to form a gate stack. Peak mobility of 132.6 cm^2^V^−1^s^−1^ was found, with three times larger current capability compared to gate oxide without La_2_O_3_, but no field oxide data were given. Figure 11 shows a summary of the discussed results.

It has to be mentioned that ternary insulators have also been investigated for MOSFET application in 4H-SiC. In particular, AlON films provided interesting and reliable results both in MOS and MOSFET applications [114,115,116]. However, ternary elements are not the focus of this review.

Very recently, Jayawardhena et al. [117] pointed out the relevance of the appropriate pre-treatment of the semiconductor to achieve reliable and stable electric characteristics by employing ALD Al_2_O_3_ films directly in contact to the bare 4H-SiC surface with no interlayers. In particular, their best results were obtained with the preparation of a nitrided surface via NO annealing, i.e., a process known to passivate surface defects, and a hydrogen exposure followed by Al_2_O_3_ deposition on the bare 4H-SiC surface [117].

A summary of the most relevant 4H-SiC MOSFETs with different high-κ gate dielectrics is reported in Table 4.

### 4.2. Binary High-κ Oxides for GaN-based MISHEMTs

Standard AlGaN/GaN MISHEMTs (see Figure 8b) are obtained by insertion of the dielectric between the metal gate and the AlGaN layer. The introduction of the gate dielectric, instead of a standard Schottky barrier gate, gives the advantage of reducing the leakage current that could limit the off-state and the gate voltage swing of the device [118]. A typical example of gate current reduction observed in HfO_2_ or CeO_2_ MISHEMTs is displayed in Figure 12a. Indeed, a gate leakage reduction of several orders of magnitude can be observed in both forward and reverse characteristics. This achievement allows a higher voltage swing in the device, which in turn results in a higher maximum drain current saturation value (I_DSmax_). Another great advantage is the very high I_ON_/I_OFF_ current ratio. Indeed, high I_ON_/I_OFF_ current ratios between 10^6^ and 10^8^ have been reported in AlGaN/GaN MISHEMTs. In Figure 12b, the I_ON_/I_OFF_ current ratio was plotted as function of the I_DSmax_. Interesting, two families of MISHEMTs can be observed depending on the leakage current level. Despite their non-outstanding I_DSmax_, some devices can exhibit very high I_ON_/I_OFF_ current ratios because of their very low leakage current. On the other hand, in other cases, despite slightly higher leakage current, extraordinary I_DSmax_ values have been demonstrated. Table 5 summarizes a survey of the most promising results obtained in normally-on AlGaN/GaN MISHEMTs using different high-κ dielectrics. Indeed, not only are Al_2_O_3_ [119,120,121] and HfO_2_ [122,123,124,125] indicated as suitable dielectrics, but many other gate oxide layers (Y_2_O_3_ [126], HZO [127], Ta_2_O_5_ [128], La_2_O_3_ [125], ZrO_2_ [129,130,131], Gd_2_O_3_ [132]) have shown promising results when integrated into GaN HEMT technology.

A relevant concern often characterizing the behaviour of high-κ binary oxides is the occurrence of charge-trapping phenomena upon bias stress [39], which can be the cause of reliability issues in GaN insulated gate transistors. Nevertheless, the electron trapping inside the Al_2_O_3_ gate insulator in GaN MISHEMTs can be used to intentionally induce a positive shift in the threshold voltage and finally obtain a normally-off operation [121]. In this context, Fiorenza et al. [133] recently studied the temperature stability of these effects, demonstrating the presence of two competitive electron trapping/de-trapping mechanisms in Al_2_O_3_ films, which were likely related to the presence of oxygen vacancies in the material.

Slightly different is the case of normally-off recessed gate hybrid MISHEMTs (see Figure 8c). In this case, the AlGaN layer below the gate region is removed, interrupting the 2DEG channel and resulting in a positive threshold voltage. The gate region is formed by a metal/oxide/GaN (MOS) interface, which requires a positive gate voltage to accumulate electrons at the oxide/GaN interface to restore the channel device. Though this approach seems to solve the crucial issue of normally-off behaviour, the complexity of these systems generates additional concerns. As an example, the lack of a 2DEG channel in the gate region causes a notable increase the channel resistance, leading to a high final on-resistance (R_ON_) and a reduced I_DSmax_. To avoid this problem, it is very important to achieve high electron mobility values [134]. Hence, the oxide/GaN interface quality is clearly a key aspect to ensure a high mobility, as are the morphology of the recessed gate region and the presence of electrically active defects [135]. In this context, the choice of dielectric gate becomes crucial. The use of SiO_2_ resulted into a poor interface quality displaying fast (interface) and slow (border) traps [136]. Dielectrics such as AlN [137], SiN [138], and their combination [139] have been also investigated as beneficial solutions to passivate surface N-vacancy, especially after recess etching damage in the gate region [140]. However, despite the good quality of the achieved interface and improved electron mobility, it was very difficult to obtain positive threshold voltages V_th_ well beyond the zero [137]. For these reasons, an increasing number of studies are focused on high-permittivity binary oxide layers for normally-off behaviour of AlGaN/GaN MISHEMTs. Table 6 shows the most promising results obtained in recessed gate hybrid MISHEMTs. ALD-deposited Al_2_O_3_ is one of the most diffused solutions for normally-off recessed gate hybrid MISHEMTs [141,142,143,144,145,146,147]. However, an excessive threshold voltage instability has been observed for Al_2_O_3_ gate insulators [121,148]. This phenomenon has been attributed to the large number of negative fixed charges incorporated in the gate stack [148,149]. As an alternative solution, ALD gate oxides with even higher dielectric constants, such as HfO_2_ [150] or ZrO_2_ [151,152,153], have been investigated for normally-off recessed MISHEMTs. Furthermore, in this case, trapped or fixed charges result in V_th_ instability issues. Other opportunities have been found in ternary oxide layers, such as HfSiO_x_ [154] or LaHfO_x_ [155].

Another important challenge in normally-off recessed gate hybrid MISHEMTs is the possibility of obtaining a very high saturation current I_DSmax_ with a well positive V_th_ value. In Figure 13, experimental values of I_DSmax_ are plotted as a function of the threshold voltage V_th_. However, the values of I_DSmax_ seem to decrease in correspondence with an increase in V_th_, thus suggesting the existence of a trade-off between a high output current and a more positive threshold voltage. In this context, a partial recession of the AlGaN barrier layer has also been explored to realize normally-off hybrid MISHEMTs. In this way, a higher 2DEG channel density is obtained. On the other hand, a more uniform and accurate AlGaN etching process is required to obtain a positive threshold voltage and normally-off devices.

Finally, to achieve normally-off behaviour in GaN-based HEMTs, the use of appropriate gate oxides with p-type semiconducting behaviour has been proposed. In fact, similarly to the most diffused p-GaN gate approach [156], the use of a p-type semiconducting oxide can lift up the conduction band at the AlGaN/GaN interface, resulting in the depletion of the 2DEG. By applying a positive gate bias V_G_, it is possible to realign the conduction band of the structures, restoring the 2DEG and the channel conduction. Among these p-type semiconducting oxides, oxides such as NiO and CuO have been taken in consideration for normally-off HEMT fabrication [157,158,159]. The origin of the p-type doping of these oxides is still debated. The existence of negatively charged Cu or Ni vacancies and the presence of interstitial oxygen [160,161] have both been considered as possible causes. Moreover, the possibility of epitaxial CVD growth on an AlGaN or GaN template makes this approach for threshold voltage engineering in GaN technology interesting [23].

**Table 6 materials-15-00830-t006:** Survey of literature data on normally-off recessed gate hybrid MISHEMTs with different high-κ dielectrics.

Dielectric	Thickness (nm)	V_TH_ (V)	Mobility (cm^2^/Vs)	R_ON_ (Ωmm or mΩcm^2^)	I_DSS_ (mA/mm)	Ref.
Al_2_O_3_	30	2	225	7.8 Ωmm	353	[141]
	38	3.5	55	27 Ωmm	336	[142]
	10	1.7	251	9.8 Ωmm	528	[143]
	20	2.9	148	7.2 Ωmm	585	[144]
	30	3.5	170	9.5 Ωmm	355	[145]
	30	2.5	192	9.6 Ωmm	620	[146]
	23	0.4	396	13.3 Ωmm	356	[147]
HfO_2_	30	1.8 (partial recessed)	876	5.2 mΩcm^2^	411	[150]
		3.6 (total recessed)	118	12.2 mΩcm^2^	146	
HfSiO_x_	15	2.2	520	10.1 Ωmm	519	[154]
LaHfO_x_	8	0.35		9.4 Ωmm	648	[144]
ZrO_2_	20	3.99	210	24 Ωmm	286	[151]
	23	2.2 (partial recessed)	850	9.2 Ωmm	590	[152]
	16	1.55 (partial recessed)	1450	7.1 Ωmm	730	[153]

## 5. Conclusions

High-permittivity binary oxides for silicon carbide (SiC) and gallium nitride (GaN) electronic devices have attracted significant interest in the last decade because of the potential benefit they can bring in the device performances. In particular, special attention has been placed on the most suitable deposition techniques for their synthesis and on their implementation in real device fabrication, in which all the processes must be compatible with industrial environments and scalable to large areas. Surely the most widely investigated binary oxide is Al_2_O_3_, as well as its combination with HfO_2_ and other materials. In fact, Al_2_O_3_ provides a good compromise among all the basic physical properties to be fulfilled by the gate dielectric for wide band gap semiconductors, namely, a dielectric constant close to that of the semiconductor, a large band gap, an appropriate band offset, a high critical electrical field, and good thermal stability. On the other hand, HfO_2_ and other oxides possess higher dielectric constants than Al_2_O_3_, but their band alignments and crystallization temperatures represent a concern in application. The most affirmed method for their synthesis has been demonstrated to be the ALD approach, which can be considered the deposition technique of choice for the fabrication of very thin films with high uniformity and conformal growth on large areas. All these capabilities render ALD as very appealing for industrial implementation. In this context, beyond the fundamental study on the impact of the deposition parameters on the films’ properties, the pre- and post-deposition conditions are relevant features for the development of a reliable high-κ technology for SiC and GaN. Cleaning treatments before high-κ thin film deposition, e.g., based on wet chemical solutions are the most suitable approach for both SiC and GaN substrates in order to limit the creation of interface defects. In spite of the “gentle” nature of the wet cleaning, interface states, as well as fixed charges within the binary oxides, still represent a great concern in practical applications. Hence, post-deposition and post-metallization annealing treatments need to be optimized in order to achieve the desired device performance. A common problem in SiC technology is the formation of an uncontrolled SiOx layer at the interface as well as residual carbon. Hence, the intentional Al_2_O_3_/SiO_2_ combination has been proposed as a possible solution, although the presence of the SiO_2_ interfacial layer partially reduces the advantage offered by the high-κ Al_2_O_3_. For that reason, the search for other material combinations and/or post-deposition treatments limiting the interfacial interaction has become mandatory. 

In regard to GaN-based devices, the implementation of Al_2_O_3_ thin films is also the most investigated and promising solution. The interaction at the interface is limited to a partial oxidation of the substrate, which in turn might be source of electrically active defects when oxynitride bonds are present. In this case, the epitaxial growth of crystalline oxides has also been widely explored as a possible route to gate insulation in GaN-based devices, considering other oxides, such as lanthanide oxides (Gd_2_O_3_, Sc_2_O_3_, and La_2_O_3_) or NiO and CeO_2_. However, the main limitations of the epitaxial oxides’ implementation are the number of structural defects occurring after the initial layers and the presence of preferential leakage current paths at the grain boundaries.

In terms of practical device application, high-κ binary oxides have already been implemented in both 4H-SiC MOSFETs and GaN-based MISHEMTs, with Al_2_O_3_ being the most widely used system. In this case, while promising results in terms of channel mobility and R_ON_ have been reported, charge-trapping effects occurring in these oxides remain a limiting factor that has to be addressed by appropriate surface preparation techniques and post-annealing conditions. In particular, the integration of high-κ oxides as gate insulators in 4H-SiC MOSFETs will require optimization of the process flow, with particular attention to the thermal budget required for ohmic contact formation, which must be compatible with the crystallization temperature of the oxide.

## Figures and Tables

**Figure 1 materials-15-00830-f001:**
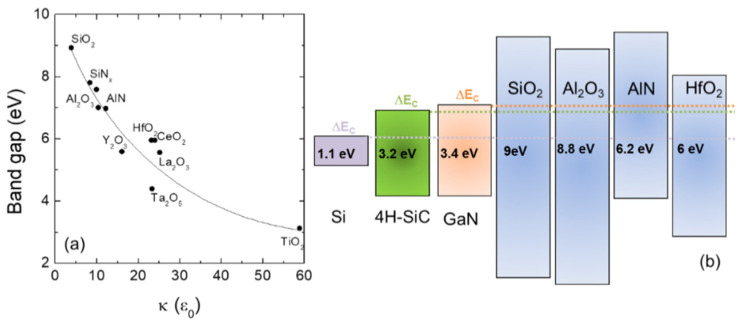
(**a**) Band gap values as a function of relative permittivity (in units of the vacuum permittivity ε_0_) for different insulators. The continuous line is a guide; (**b**) schematic illustration (in scale) of the band alignments of some common insulators with the semiconductor materials under consideration (i.e., silicon, 4H-SiC, and GaN). The light purple, green, and orange dotted lines indicate the conduction band edge of the Si, 4H-SiC, and GaN semiconductors, respectively.

**Figure 2 materials-15-00830-f002:**
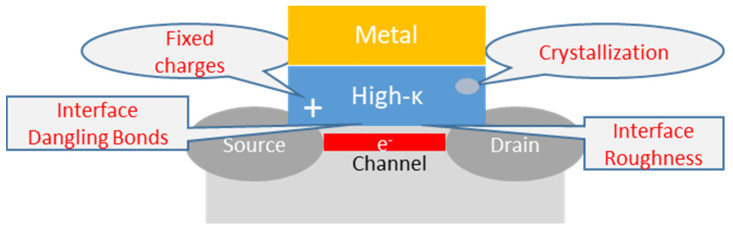
Schematic representation of the main issues affecting the functionality of high-κ binary gate oxides in a transistor.

**Figure 3 materials-15-00830-f003:**
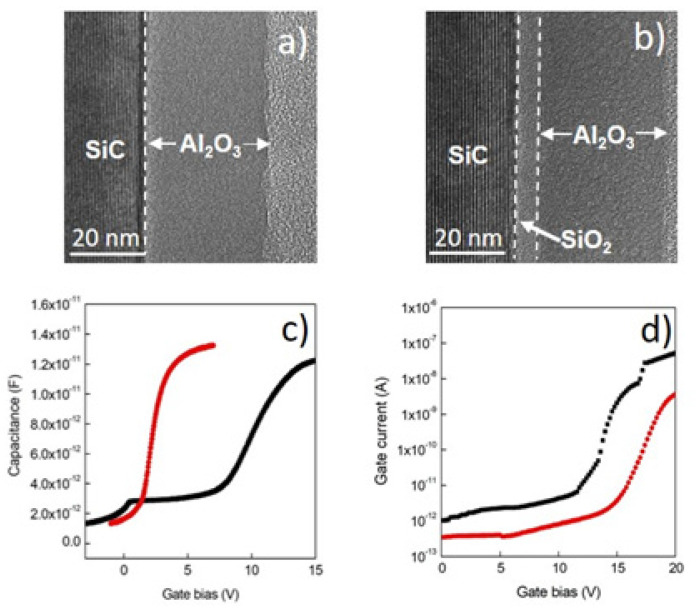
TEM images of Al_2_O_3_ thin films grown by PE-ALD on 4H-SiC (**a**) and SiO_2_/4H-SiC (**b**) substrates and their relative electrical characteristics in terms of C-V curves (**c**) and I-V measurements (**d**) performed on MOS capacitors. Black and red lines are related to Al_2_O_3_ thin films deposited on SiO_2_/4H-SiC and 4H-SiC substrates, respectively. Reproduced with permission from [40]. Copyright © 2016 WILEY-VCH Verlag GmbH & Co. KGaA.

**Figure 4 materials-15-00830-f004:**
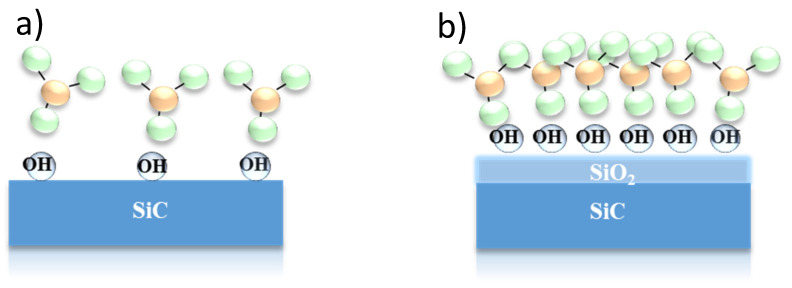
Schematic representation of the chemical impact of the different substrate surfaces on the Al_2_O_3_ nucleation processes, in the case of a bare SiC substrate (**a**) or a SiC substrate with a thin SiO_2_ layer on the top (**b**). Reproduced with permission from [40]. Copyright © 2016 WILEY-VCH Verlag GmbH & Co. KGaA.

**Figure 5 materials-15-00830-f005:**
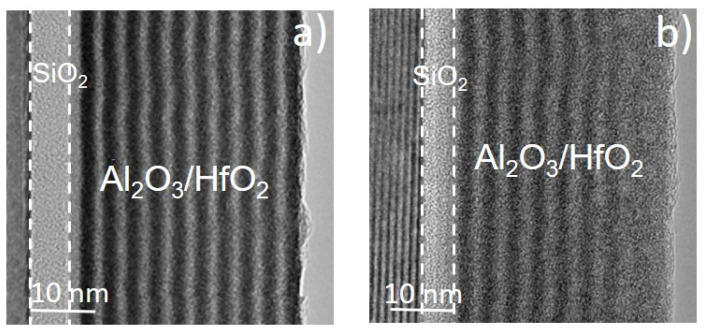
TEM image of (**a**) as deposited and (**b**) 800 °C annealed Al_2_O_3_/HfO_2_ nanolaminate, deposited onto SiO_2_/SiC substrate. Reproduced from [63]. Copyright © 2020 Authors.

**Figure 6 materials-15-00830-f006:**
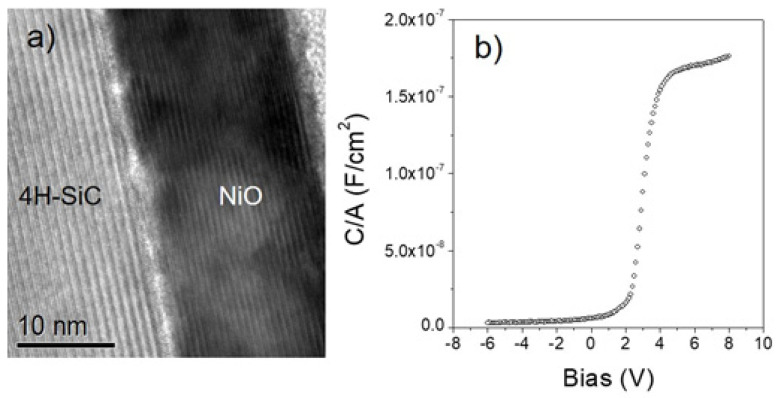
(**a**) High-resolution cross-section TEM image of a NiO film deposited by MOCVD on 4H-SiC at 500 °C; (**b**) C-V curve acquired on a NiO/4H-SiC MOS capacitor. Reproduced with permission from [81]. Copyright © 2013 Elsevier Ltd.

**Figure 7 materials-15-00830-f007:**
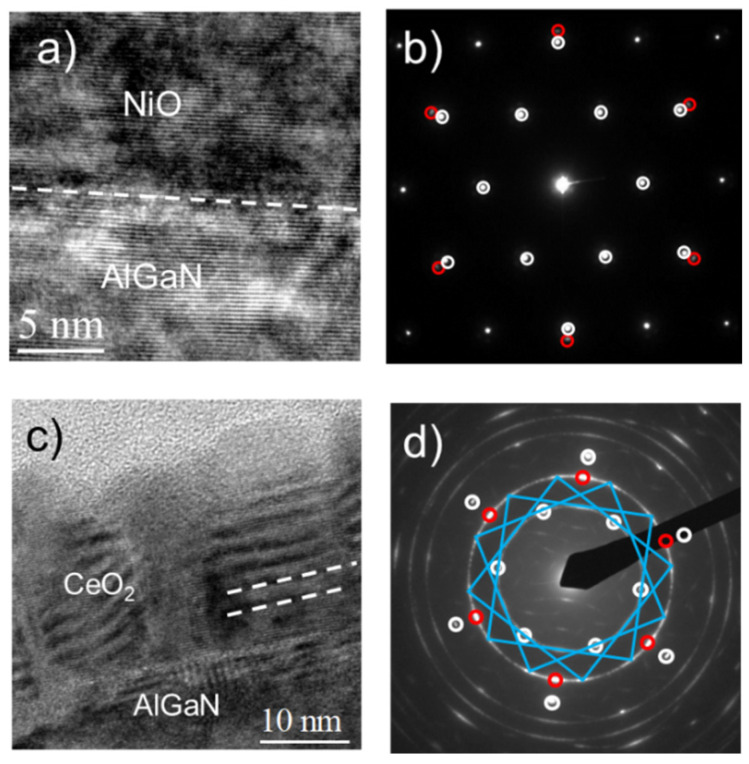
High-magnification cross-section TEM images (**a**) and in-plane SAED patterns (**b**) of NiO thin film deposited by MOCVD on AlGaN/GaN heterostructure at 500 °C. High-magnification cross-section TEM image (**c**) and in-plane SAED patterns (**d**) of CeO_2_ thin film deposited by MOCVD on AlGaN/GaN heterostructure at 500 °C. Panel (**a**): reproduced with permission from [23]. Copyright © 2012 AIP Publishing; Panel (**d**): reproduced with permission from [25]. Copyright © 2013 AIP Publishing.

**Figure 8 materials-15-00830-f008:**
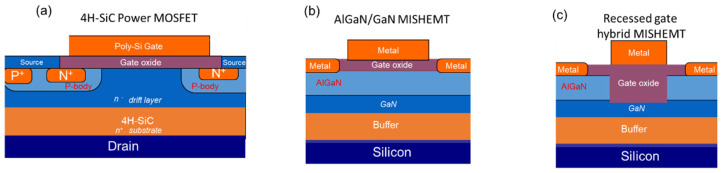
Schematic cross section of (**a**) a 4H-SiC power MOSFET, (**b**) an AlGaN/GaN MISHEMT, and (**c**) a recessed gate hybrid MISHEMT.

**Figure 9 materials-15-00830-f009:**
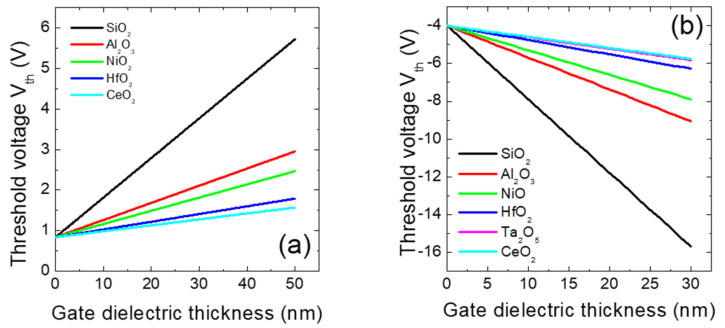
Calculated threshold voltage of SiC MOSFETs (**a**) and AlGaN/GaN MISHEMTs (**b**) as a function of the gate insulator layer thickness for different high-κ materials. Panel (**b**): reproduced with permission from [100]. Copyright © 2014 WILEY-VCH Verlag GmbH & Co. KGaA.

**Figure 10 materials-15-00830-f010:**
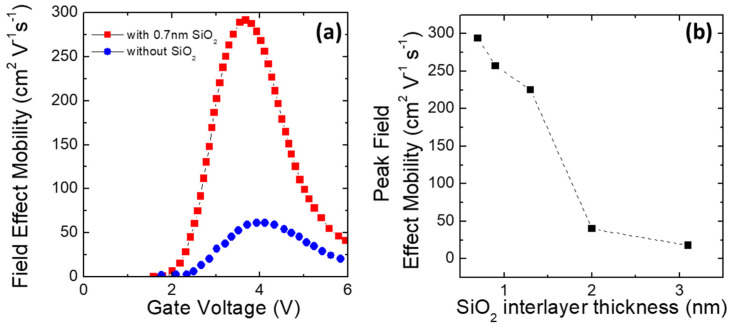
(**a**) Comparison between the field-effect mobility obtained in 4H-MOSFETs fabricated using Al_2_O_3_ insulators with and without an ultrathin thermally grown SiOx layer inserted between the Al_2_O_3_ and SiC interface. (**b**) Peak value of the field-effect mobility obtained using SiOx layers with different thicknesses. The data are taken from [107,108].

**Figure 11 materials-15-00830-f011:**
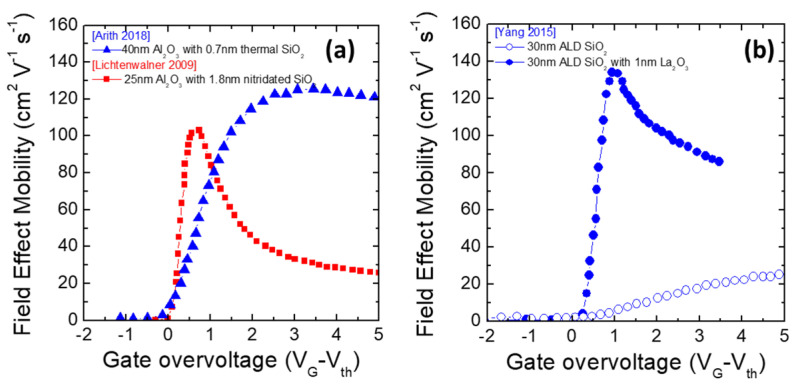
(**a**) Comparison between the field-effect mobility obtained in 4H-MOSFETs fabricated using Al_2_O_3_ insulators with the insertion of ultrathin thermally grown or nitridated SiO_2_ layers. (**b**) Comparison between the field-effect mobility obtained in 4H-SiC MOSFETs fabricated using SiO_2_ insulators with the insertion of ultrathin La_2_O_3_ layer. The data are taken from Refs. [43,111,113].

**Figure 12 materials-15-00830-f012:**
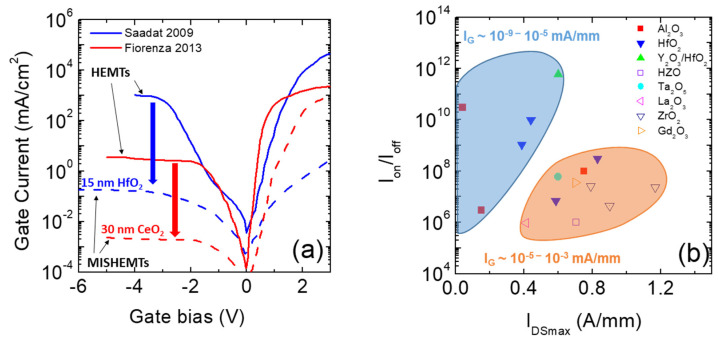
(**a**) Comparison of the gate current–voltage characteristics of AlGaN/GaN HEMTs (Schottky gate) and MISHEMTs employing HfO_2_ and CeO_2_ gate insulators. The data are taken from [25,118]. (**b**) I_ON_/I_OFF_ versus I_DSmax_ for MISHEMTs using different gate oxides. The data are taken from Table 6 and references therein.

**Figure 13 materials-15-00830-f013:**
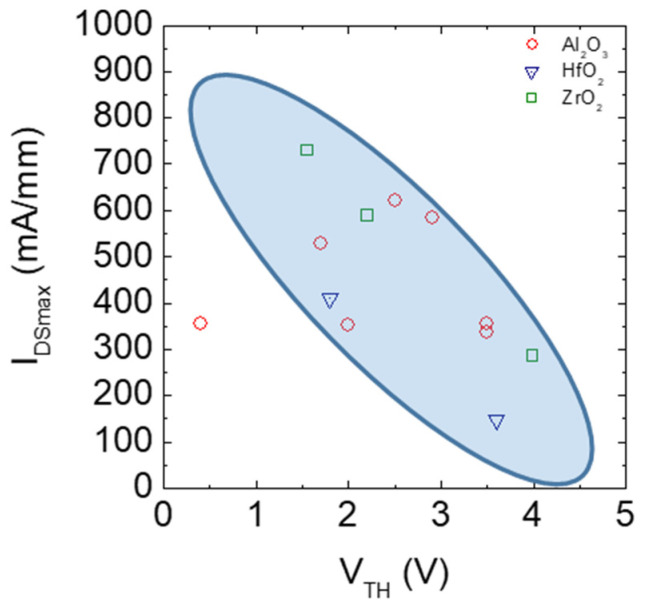
I_DSmax_ versus V_th_ value for recessed hybrid MISHEMTs using different high-κ binary gate oxides. The data are taken from [141,142,143,144,145,146,147,150,151,152,153].

**Table 1 materials-15-00830-t001:** Principal physical properties of high-κ gate binary oxides.

Oxide	Dielectric Constant	Band Gap (eV)	Crystallization Temperature	Ref
Al_2_O_3_	10	9	900 °C	[17,18]
HfO_2_	~20	5.6–5.8	500 °C	[17,18,22]
NiO	11.7	4	300 °C	[23,24]
CeO_2_	26	6	500 °C	[25]
Sc_2_O_3_	12–14	6.0	>400 °C	[26,27,28,29]
Y_2_O_3_	10	5.5	>400 °C	[26,28,29]
Gd_2_O_3_	~20	5.0–5.45	>400 °C	[26,28]
La_2_O_3_	~20	5.4–5.6	>400 °C	[17,18,27,29]
ZrO_2_	25	5.8	>400 °C	[17,18]
Ga_2_O_3_	~10	5	>500 °C	[30]

**Table 2 materials-15-00830-t002:** Comparison of the main features of the common deposition techniques for high-κ oxides in microelectronics [29,31,32].

	ALD	MBE	CVD	PVD
**Thickness range**	≤2000 Å	≤2000 Å	≥100 Å	≥100 Å
**Deposition rate**	Low 1–5 nm/min	High 0.01–0.3 µm/min	High 1–10 µm/h	Medium 0.1–1 µm/h
**Step coverage** **Aspect ratio**	100% 60:1	25–50% 1:1	70% 1:1	25–50% 1:1
**Deposition** **temperature**	25–400 °C	500–1000 °C	300–1100 °C	200–500 °C
**Film type** **availability**	High (limited for metals)	High (limited for Metals)	High (limited for metals)	High for metals and conductive materials

**Table 3 materials-15-00830-t003:** Physical and structural properties of high-κ oxides epitaxially grown on GaN.

Oxide	Dielectric Constant	Lattice Constant (Å)	Mismatch to (0001) GaN (%)	Deposition Technique	Ref.
Gd_2_O_3_	9	10.813	20.1	MBE	[82]
Sc_2_O_3_	13–14	9.845	9.2	PVD and MBE	[83,84]
La_2_O_3_	18–27	4.211	6.5	MBE	[83]
CeO_2_	15–26	5.411	6	MOCVD	[25,81]
NiO	11.9	4.177	5	Thermal oxidation or MOCVD	[23,24,81]

**Table 4 materials-15-00830-t004:** Survey of literature data on 4H-SiC MOSFETs with different high-κ gate dielectrics.

Gate Insulator	Thickness (nm)	V_th_ (V)	µ_FE_ (cm^2^V^−1^s^−1^)	D_it_ (cm^−2^eV) at E_C_ − E_t_ = 0.2 eV	Ref.
Al_2_O_3_	35	2.8	64	8 × 10^11^	[108]
Al_2_O_3_	33	0.5 -3	52	1 × 10^11^ cm^−2^ (integral)	[117]
Al_2_O_3_ on SiO_2_	35 + 2	2.8	18	8 × 10^11^	[108]
	35 + 0.7	2.8	300	5 × 10^11^	[108]
	40 + 0.7	2	120	6 × 10^11^	[111]
	25 + 1.8	0.8	106	-	[43]
SiO_2_ on La_2_O_3_	30 + 1	3	132	-	[113]
AlON	60 + 10	> 0	26.9	1 × 10^11^	[115]

**Table 5 materials-15-00830-t005:** Survey of literature data on normally-on AlGaN/GaN MISHEMTs with different high-κ gate dielectrics.

Dielectric	Thickness (nm)	V_th_ (V)	I_Dmax_ (mA/mm)	I_G-leak_ (mA/mm)	I_ON_/I_OFF_	Ref.
Al_2_O_3_	25	−7.0	150	5.0 × 10^−5^	3.0 × 10^6^	[119]
	15	−7.0	750	8.0 × 10^−5^	1.0 × 10^8^	[120]
	30	−8.0	40	1.0 × 10^−8^	3.0 × 10^10^	[121]
HfO_2_	20	−1.1	440	2.2 × 10^−7^	1.0 × 10^10^	[122]
	12	−8.0	386	1.1 × 10^−9^	1.1 × 10^9^	[123]
	23	−6.0	830	3.0 × 10^−6^	3.0 × 10^8^	[124]
	8	−3.7	585	6.5 × 10^−5^	6.9 × 10^6^	[125]
Y_2_O_3_/HfO_2_	1/12	−5.0	600	3.0 × 10^−9^	6.0 × 10^11^	[126]
Ta_2_O_5_	24	−9.7	600	1.0 × 10^−5^	6.0 × 10^7^	[128]
La_2_O_3_	8	−2.9	409	1.0 × 10^−4^	9.7 × 10^5^	[125]
ZrO_2_	30	−7	1168	5.4 × 10^−4^	2.3 × 10^7^	[129]
	10	−4.2	900	2.0 × 10^−4^	4.5 × 10^6^	[130]
	10	−3.9	790	3.0 × 10^−5^	2.6 × 10^7^	[131]
HfZrO_x_	20	−12	705	6.0 × 10^−4^	1.0 × 10^7^	[127]
Gd_2_O_3_	4	−6.5	700	1.0 × 10^−6^	3.5 × 10^7^	[132]

## Data Availability

The data that support the findings of this study are available from the corresponding author upon reasonable request.

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
