# Peer review of "Structural and Insulating Behaviour of High-Permittivity Binary Oxide Thin Films for Silicon Carbide and Gallium Nitride Electronic Devices"

_materials, 2022, doi:10.3390/ma15030830_

Round 1
Reviewer 1 Report
This review by R. L. Nigro et al reports on the structural and dielectric properties of high-k binary oxides. It is such a comprehensive overview on high permittivity binary oxide thin films for post-Si electronic applications. However, there are several issues to be addressed before accepting the manuscript for publication in the Materials.
Comments:
- The scope and content of the review is about structural and dielectric features of high-permittivity oxides. However, the title is very general as if it discusses other physical properties such as optical, thermal, mechanical, etc. I wonder why the authors are afraid of including other physical properties of high-k binary oxides. Thus, I suggest the authors either to specify the title or include other physical properties of high-k binary oxides.
- The opening paragraph of introduction section has no citation of the historical developments in microelectronic devices. The same is true for most paragraphs in page 4 and page 5. That needs to be corrected.
- The color bar in Figure 1(b) needs specification/description in the caption. Moreover, the labelling and caption of Figure 3(a) & (b) is also not clear.
- At the opening of Table 2, the deposition methods (horizontal/row) and parameters (vertical/column) are represented by one heading/title as “Method”. However, the methods and parameters should be separate. The authors can replace “Method” by “Parameters” and the put the “Method” just above the listed deposition methods listed. Furthermore, there is unnecessary and empty row in Table 6.
- The manuscript is full of grammatical errors and types. For example, the last sentences of abstract are written in future tense like this: “The impact of deposition modes and pre- or post-deposition treatments will be discussed. Moreover, the dielectric behaviour of these films will be presented, reporting some examples of high-κ binary oxides applied to SiC and GaN transistors” (line 22 – 24). Moreover, there are a number of grammatical errors for example “However, up to date, several issues remain objects of investigation” (page 6, line 190), “with respect” (page, line 292 and line 295); or full of typos such as “different substrate surface on the on the Al2O3” (line 282), etc.
- I could not find citation of prominent works published on related topics, for example, the detailed review on the thermal, mechanical, electrical, optical, and structural properties of high-k dielectrics by John T. Gaskins et al 2017 ECS J. Solid State Sci. Technol. 6 N189.

Author Response
Reviewer#1
This review by R. L. Nigro et al reports on the structural and dielectric properties of high-k binary oxides. It is such a comprehensive overview on high permittivity binary oxide thin films for post-Si electronic applications. However, there are several issues to be addressed before accepting the manuscript for publication in the Materials.
We thank the reviewer for the positive comment concerning the applicative aspect of our work. We revised the manuscript addressing point-by-point his/her suggestions. All changes have been highlighted in the revised version of the manuscript and figure 3 has been modified as requested.
- The scope and content of the review is about structural and dielectric features of high-permittivity oxides. However, the title is very general as if it discusses other physical properties such as optical, thermal, mechanical, etc. I wonder why the authors are afraid of including other physical properties of high-k binary oxides. Thus, I suggest the authors either to specify the title or include other physical properties of high-k binary oxides.
We agree with the referee on the wide potentiality of binary oxides because of their large number of physical and chemical properties. Nevertheless, in order to have a defined focus on the aim of the review, the title has been changed as “Structural and insulating behaviour of high permittivity binary oxides thin films for silicon carbide and gallium nitride electronic devices”. The aim of the review is to address the potentiality of binary oxides as gate insulator for wide band gap semiconductors, as SiC and GaN substrates.
- The opening paragraph of introduction section has no citation of the historical developments in microelectronic devices. The same is true for most paragraphs in page 4 and page 5. That needs to be corrected.
Several references have been added from page 1 to page 5, as suggested by the referee.
- The color bar in Figure 1(b) needs specification/description in the caption. Moreover, the labelling and caption of Figure 3(a) & (b) is also not clear.
Figure 3 has been modified in order to make the markers more visible. The captions of figures 1 and 3 have been also corrected.
- At the opening of Table 2, the deposition methods (horizontal/row) and parameters (vertical/column) are represented by one heading/title as “Method”. However, the methods and parameters should be separate. The authors can replace “Method” by “Parameters” and the put the “Method” just above the listed deposition methods listed. Furthermore, there is unnecessary and empty row in Table 6.
Tables 2 and 6 have been modified. In particular, “Method” has been cancelled and the Table2 appears now clear, while in Table 6 the unnecessary row has been eliminated.
- The manuscript is full of grammatical errors and types. For example, the last sentences of abstract are written in future tense like this: “The impact of deposition modes and pre- or post-deposition treatments will be discussed. Moreover, the dielectric behaviour of these films will be presented, reporting some examples of high-κ binary oxides applied to SiC and GaN transistors” (line 22 – 24). Moreover, there are a number of grammatical errors for example “However, up to date, several issues remain objects of investigation” (page 6, line 190), “with respect” (page, line 292 and line 295); or full of typos such as “different substrate surface on the on the Al2O3” (line 282), etc.
The paper has been revised and all the typos and grammatical errors have been corrected.
- I could not find citation of prominent works published on related topics, for example, the detailed review on the thermal, mechanical, electrical, optical, and structural properties of high-k dielectrics by John T. Gaskins et al 2017 ECS J. Solid State Sci. Technol. 6 N189.
The interesting reference suggested by the referee has been added as Ref 20.

Reviewer 2 Report
The manuscript summarized the development of binary oxide thin films for the electronics. I recommend the publication of the review after a minor revision. The details are listed below:
- Some binary oxides are absent in the introduction, such as ZrO2. Some newly born films should also be involved in this review, including Ga2O3, In2O3, and SnO2. Moreover, their properties should also be introduced in Table 1.
- Some references should be added, including Chem. Rev., 2018, 118, 6236; Nat. Commun., 2020, 11, 2502; Science, 2017, 358, 332.
- The format of references in Table 1, 4, 5, 6, and 7 are different. Moreover, where is Table 3?
Author Response
Reviewer#2
The manuscript summarized the development of binary oxide thin films for the electronics. I recommend the publication of the review after a minor revision.
We thank the reviewer his/her suggestions. All changes have been highlighted in the revised version of the manuscript.
- Some binary oxides are absent in the introduction, such as ZrO2. Some newly born films should also be involved in this review, including Ga2O3, In2O3, and SnO2. Moreover, their properties should also be introduced in Table 1.
We agree with the referee on the importance of ZrO2 as insulating oxide, in fact several decades ago it was widely considered for Si-based technology. To our best knowledge, however, very few attempts have been carried out on SiC and/or GaN substrates, and the results seemed to be quite poor. We included the ZrO2 and Ga2O3 in the list (Table 3) of potentially appealing materials to be used as gate oxides. In regards to the In2O3 and SnO2, despite they are interesting materials for the recent development of applications based on WBG semiconductors, they are mainly used for physical properties different from their insulating behavior. In order to avoid such kind of ambiguity, we changed the title of the paper as “Structural and insulating behaviour of high permittivity binary oxides thin films for silicon carbide and gallium nitride electronic devices”. So that it could be clear that the insulating properties of the materials is the main physical characteristic which has been considered and discussed.
- Some references should be added, including Chem. Rev., 2018, 118, 6236; Nat. Commun., 2020, 11, 2502; Science, 2017, 358, 332.
The references suggested by the referee are very interesting but they are not strictly related to the topic of the present manuscript. We will consider them for future work.
- The format of references in Table 1, 4, 5, 6, and 7 are different. Moreover, where is Table 3?
The reviewer is perfectly right, the table 3 is missing because in our intention it was the table 4. We are sorry for this inconvenience and we re-numbered all the tables and now all of them possess the same format.

Reviewer 3 Report
Post-Si electronic device might include but not limit to SiC, germanium, gallium nitride, 1D nanowires/tubes, 2D graphene, and other dichalcogenide materials and ferroelectrics. Since the authors limit their review to the WBG semiconductor devices (such as GaN and SiC), the paper title should be consistent with this limited topic.
On the other hand, this review paper is written very well, and can be published as its current form.
Author Response
Reviewer#3
Post-Si electronic device might include but not limit to SiC, germanium, gallium nitride, 1D nanowires/tubes, 2D graphene, and other dichalcogenide materials and ferroelectrics. Since the authors limit their review to the WBG semiconductor devices (such as GaN and SiC), the paper title should be consistent with this limited topic. On the other hand, this review paper is written very well, and can be published as its current form.
We thank you very much the reviewer4 for the nice evaluation of our paper, and we agree with his/her suggestion on the need to modify the title, which has been changed as “Structural and insulating behaviour of high permittivity binary oxides thin films for silicon carbide and gallium nitride electronic devices”.

Reviewer 4 Report
Authors address a hot topic related to high power and high frequency transistors based on WBG semiconductors SiC and GaN with target on high permittivity binary oxides (amorphous and crystalline) as gate dielectric layers. The review is comprehensive, surveys the literature, highlights the requirements for the ideal gate dielectric and addresses the issues for developing devices with such gate oxides. Al2O3 (without or with thermally grown SiO2) is the most studied oxide, showing high dielectric constant, large band gap, appropriate band off-sets, high critical electrical field and good thermal stability, and with established ALD growth. Reports on other oxides such as HfO2, NiO, CeO2, Gd2O3, Sc2O3 or Al2O3/HfO2 multilayer laminated systems are also reviewed. The most promising deposition technique is (PE-)ALD with atomic level accuracy. Careful attention is given to issues related to reducing density of interface defects, or for the case of epitaxial oxides to structural defects occurring after initial layers growth. A problem related to high-k oxides is the density of electronic defects that can be reduced by thermal treatments or by optimizing deposition processes. Other problem that represents a prerequisite is the quality of WBG semiconductor surface, in SiC – reducing interface states density, in GaN-based materials - presence of a large concentration of defects (nitrogen vacancies, structural/morphological imperfections, residual contaminations). Authors discuss pre-deposition surface cleaning and treatment procedures of WBG semiconductor-based surfaces, annealing procedures before or after gate dielectric deposition. Related to devices, charge trapping effects in oxides remain a limiting factor, and best gate-processing conditions for MOSc-HEMTs should be found.
I recommend the paper for publication in Materials.
I have very minor comments:
- Please explain caption in Figure3c,d – red and black curves.
- Please revised whole text, different typos: line 96 – eV instead of V units; line 273 – should be relative permittivity etc.
- Please revise Conclusions – many typos.
- Could you detail some guidance/ideas of future developments of reviewed topic?
What about epitaxial NiO that showed high quality epitaxial growth to be integrated in transistors?
The reduction of fixed charges how can be further tackled for better device performance?
- References – complete formatting is necessary.
- Other suggested references:
- Sabria Benrabah, Maxime Legallais, Pascal Besson, Simon Ruel, Laura Vauche, Bernard Pelissier, Chloé Thieuleux, Bassem Salem, Matthew Charles. H3PO4-based wet chemical etching for recovery of dry-etched GaN surfaces. Applied Surface Science 2021, 28 , 152309. https://doi.org/10.1016/j.apsusc.2021.152309
- Laura Vauche, Antoine Chanuel, Eugénie Martinez, Marie-Christine Roure, Cyrille Le Royer, Stéphane Bécu, Romain Gwoziecki, Marc Plissonnier, Study of an Al2O3/GaN Interface for Normally Off MOS-Channel High-Electron-Mobility Transistors Using XPS Characterization: The Impact of Wet Surface Treatment on Threshold Voltage VTH. ACS Appl. Electron. Mater. 2021, 3, 3, 1170–1177. https://doi.org/10.1021/acsaelm.0c01023

Author Response
Reviewer#4
Authors address a hot topic related to high power and high frequency transistors based on WBG semiconductors SiC and GaN with target on high permittivity binary oxides (amorphous and crystalline) as gate dielectric layers. The review is comprehensive, surveys the literature, highlights the requirements for the ideal gate dielectric and addresses the issues for developing devices with such gate oxides. Al2O3 (without or with thermally grown SiO2) is the most studied oxide, showing high dielectric constant, large band gap, appropriate band off-sets, high critical electrical field and good thermal stability, and with established ALD growth. Reports on other oxides such as HfO2, NiO, CeO2, Gd2O3, Sc2O3 or Al2O3/HfO2 multilayer laminated systems are also reviewed. The most promising deposition technique is (PE-)ALD with atomic level accuracy. Careful attention is given to issues related to reducing density of interface defects, or for the case of epitaxial oxides to structural defects occurring after initial layers growth. A problem related to high-k oxides is the density of electronic defects that can be reduced by thermal treatments or by optimizing deposition processes. Other problem that represents a prerequisite is the quality of WBG semiconductor surface, in SiC – reducing interface states density, in GaN-based materials - presence of a large concentration of defects (nitrogen vacancies, structural/morphological imperfections, residual contaminations). Authors discuss pre-deposition surface cleaning and treatment procedures of WBG semiconductor-based surfaces, annealing procedures before or after gate dielectric deposition. Related to devices, charge trapping effects in oxides remain a limiting factor, and best gate-processing conditions for MOSc-HEMTs should be found.
I recommend the paper for publication in Materials.
We thank very much the referee for his/her very positive comment on our manuscript. We addressed all his/her suggestions as described in the following:
- Please explain caption in Figure3c,d – red and black curves.
The Figure 3 caption has been modified in order to make clearer the meaning of the red and black curves.
- Please revised whole text, different typos: line 96 – eV instead of V units; line 273 – should be relative permittivity etc
We thank the referee for the suggestions. The paper has been revised and all the typos and grammatical errors have been corrected.
2. Please revise Conclusions – many typos.
The paper has been revised and all the typos and grammatical errors have been corrected.
3. Could you detail some guidance/ideas of future developments of reviewed topic?
The conclusion section has been revised in order to describe some ideas on future developments. In particular, the following part has been added:
“In spite of the “gentle” nature of the wet cleaning, interface states, as well as fixed charges within these binary oxides, represent still a great concern in practical applications. Hence, post-deposition and post-metallization annealing treatments need to be optimized in order to achieve the desired devices performances. A common problem in SiC technology is the formation on uncontrolled SiOx layer at the interface as well as residual carbon. Hence, the intentional Al2O3/SiO2 combination has been proposed as a possible solution, although the presence of the SiO2 interfacial layer partially reduces the advantage offered by the high-k Al2O3. For that reason, the search of other materials combination and/or post-deposition treatments limiting the interfacial interaction has become mandatory. In regards to GaN-based devices, the implementation of Al2O3 thin films is also the most investigated and promising solution. The interaction at the interface is limited to a partial oxidation of the substrate, which in turn might be source of electrically active defects, when oxynitride bonds are present. In this case, also the epitaxial growth of crystalline oxides has been largely explored as possible route to gate insulation in GaN-based devices, considering other oxides, such as lanthanide oxides (Gd2O3, Sc2O3 and La2O3) or NiO and CeO2. However, one of the main limitations of the epitaxial oxides implementation is the amount of structural defects occurring after the initial layers, as well as the presence of preferential leakage current paths at the grain boundaries.
In terms of practical devices application, high-k binary oxides have been already implemented in both 4H-SiC MOSFETs and GaN-based MISHEMTs, being the Al2O3 the most widely used system. In this case, while promising results in terms of channel mobility and RON have been reported, charge trapping effects occurring in these oxides remain a limiting factor, which has to be addressed by appropriate surface preparation techniques and post-annealing conditions. In particular, the integration of high-k oxides as gate insulators in 4H-SiC MOSFETs will require an optimization of the process flow, with a particular attention to the thermal budget required for ohmic contact formation, which must be compatible with the crystallization temperature of the oxide.”
What about epitaxial NiO that showed high quality epitaxial growth to be integrated in transistors?
While (111)NiO thin insulating layers seem to be an appealing choice as epitaxial gate oxide, its integration in real transistor has not been attempted yet.
The reduction of fixed charges how can be further tackled for better device performance?
Certainly the reduction of the fixed charged is one of the principle problem to limit probably by appropriate surface preparation techniques and post-annealing conditions. In particular, the integration of high-k oxides as gate insulators in 4H-SiC MOSFETs will require an optimization of the process flow, with a particular attention to the thermal budget required for ohmic contact formation, which must be compatible with the crystallization temperature of the oxide.
4. References – complete formatting is necessary.
All the references have been formatted
5. Other suggested references:
- Sabria Benrabah, Maxime Legallais, Pascal Besson, Simon Ruel, Laura Vauche, Bernard Pelissier, Chloé Thieuleux, Bassem Salem, Matthew Charles.H3PO4-based wet chemical etching for recovery of dry-etched GaN surfaces. Applied Surface Science2021,28, 152309.https://doi.org/10.1016/j.apsusc.2021.152309
- Laura Vauche, Antoine Chanuel, Eugénie Martinez, Marie-Christine Roure, Cyrille Le Royer, Stéphane Bécu, Romain Gwoziecki, Marc Plissonnier, Study of an Al2O3/GaN Interface for Normally Off MOS-Channel High-Electron-Mobility Transistors Using XPS Characterization: The Impact of Wet Surface Treatment on Threshold Voltage VTH. ACS Appl. Electron. Mater. 2021, 3, 3, 1170–1177. https://doi.org/10.1021/acsaelm.0c01023
We thank the referee for the suggested references, they are interesting and have been added as refs 77 and 78.
